# Analysis of Operational Efficiency and Cost Differences between Local and General Anesthesia for Vitreoretinal Surgery

**DOI:** 10.3390/healthcare10101918

**Published:** 2022-09-30

**Authors:** Mohammad Z. Siddiqui, Muhammad Z. Chauhan, Alvin F. Stewart, Ahmed B. Sallam

**Affiliations:** 1Department of Ophthalmology, Harvey and Bernice Jones Eye Institute, University of Arkansas for Medical Sciences, Little Rock, AR 72205, USA; 2Department of Anesthesiology, University of Arkansas for Medical Sciences, Little Rock, AR 72205, USA

**Keywords:** vitreoretinal surgery, local anesthesia, general anesthesia, cost analysis

## Abstract

There has been a growing trend of using local anesthesia (LA) compared to general anesthesia (GA) over the last two decades in VR surgery. We aim to answer the following question: what is the institutional benefit of LA versus GA use in operation-room time, anesthesia duration, earlier discharge from an outpatient surgery facility, and the estimated cost savings? We conducted a retrospective analysis of 1476 eyes that underwent vitreoretinal surgery over a 6-year period from a single site; 61.8% of patients received GA and 38.2% received LA for VR surgery. Anesthesia, surgical, and recovery times were significantly shorter with LA (100.49, 66.47, 66.47 mins) vs. GA (145.53, 100.14, 75.08 mins). Anesthesia, surgical, and recovery costs were significantly lower for eyes that received LA, with an estimated adjusted cost reduction of USD 1516 per surgery using LA instead of GA. Use of LA for vitreoretinal surgery is associated with better operational efficiency, earlier patient discharge, and significant cost reduction.

## 1. Introduction

The number of vitreoretinal (VR) procedures performed in the United States has steadily increased over the past several years, growing from 527,050 procedures in 2000 to more than 3 million procedures in 2014, with the majority of surgeries performed on a one-day (outpatient) surgery basis [1]. Over the last two decades, there has also been an increase in the use of LA for VR surgery in Europe and the USA [2,3]. Monitored anesthesia care with local anesthesia (LA) is effective for many VR surgeries, and it is associated with a comparable rate of perioperative outcomes and surgical complications when compared to general anesthesia (GA) [1,3,4,5,6]. There are several advantages to using LA over GA, including that LA poses less cardiovascular risks, is an option for emergency cases in non-fasting patients, has a shorter recovery duration, and allows an earlier patient discharge [3,5,7,8].

The rising cost of health care has necessitated exploring cost utilization strategies. One prior study assessing the cost of a vitrectomy performed under LA versus GA showed that using LA can reduce the total cost burden of the surgery by 46.6% in comparison to GA at two tertiary-care centers in Indonesia [8]. For other outpatient surgeries, such as a thyroidectomy in the United States, Snyder et al. saw that LA reduced the postoperative time spent in an outpatient surgery setting with cost savings compared to GA [9]. Another group found similar results for outpatient hand surgery, showing that a local block achieved substantial cost savings in complex hand surgeries compared to GA [10]. To date, no study has analyzed the operational efficiency and cost reduction in LA versus GA use for vitreoretinal surgery in the United States, and the literature is very scarce worldwide [8]. 

With the previously mentioned limitations of the current literature, we conducted this study to quantify the institutional benefit of using LA versus GA for vitreoretinal surgery. Specifically, we analyzed the operation-room time, anesthesia duration, and time to discharge from the one-day surgery facility. We secondarily compared the cost of the operating room and postoperative anesthesia care unit (PACU) utilization between different methods of anesthesia delivery. 

## 2. Methods

We extracted data from the electronic medical record (EMR) system of the University of Arkansas for Medical Sciences (UAMS), a single-site tertiary-care facility. This study was conducted in accordance with the Declaration of Helsinki. An institutional board review exemption was issued by the university, and no patient consent was required. 

To provide a large data cohort, we selected a study period from January 2015 to September 2021. We focused on a cohort that was 18 years of age or older. We identified an initial database of a total of 2065 eyes that received one-day vitreoretinal surgery. We cleaned the data through a stepwise process that started by organizing data by the date of the first surgery and then filtering out multiple surgeries per eye in each patient. This allowed us to represent only the first surgery in the final cleaned analytical dataset. The final dataset included 1473 eyes, with 635 (43.11%) eyes receiving LA and 838 (56.89%) eyes receiving GA.

We collected data pertaining to age, sex, ethnicity, pre-existing systemic comorbidities, body mass index (BMI), smoking status, type of anesthesia administered, type of vitreoretinal procedure, anesthesia time, surgery time, and duration of PACU stay. 

For analysis, we grouped the data into five categories based on the type of retina surgery (as determined by surgical CPT codes and ‘Procedure Text’ extract): scleral buckle for retinal detachment (67107), pars plana vitrectomy (PPV) for retinal detachment (67108), PPV plus a scleral buckle for retinal detachment (67108), noncomplex PPV including macular surgery, vitreous hemorrhage and opacities, dropped nucleus (67041, 67042, 67043, 67036, 67015, 67030, 67031, 67039, 67040, and 67121), and complex PPV (67113). We excluded analysis from extraocular VR procedures that are occasionally done in the operating room and procedures that we could not fit into one of the previous five categories, including the following CPT codes: 67005, 67010, 67027, 67028, 67145, 67141, 67220, 67208, 67210, 67221, 67225, 67227, 67228, 67229, 67550, 67515, 67025, 67101, 67105, 67120, 67250, 67225, 67115, and 67218. 

The major outcomes of interest in this study were the operating room, anesthesia, and PACU times and the cost for vitreoretinal surgical procedures operated under local anesthesia compared to general anesthesia. Operating-room time was defined from the moment the ophthalmologist makes the initial invasive incision, and this is communicated to the operating-room circulator who documents the procedure start time. Operating-room time ends when the ophthalmologist has made the final closure of the procedure site, which is again documented by the operating-room circulator. Anesthesia time is defined as the time from when the anesthesia provider enters the operating room with the patient and concludes after procedure completion and when the patient has been delivered to the PACU. PACU time was defined as the time the patient has been handed off to the recovery room to the final discharge time from the ambulatory surgery center. Cost was estimated through time-driven activity-based costing (TDABC). We calculated the costs of the operating room and PACU with two different equations that included an initial base charge (USD 2795.00 and USD 838.72, respectively) plus an additional time charge in 15-minute blocks (USD 616.00 and USD 125.00, respectively). Anesthesia cost was calculated by a base unit from the CPT code plus modifying units based upon the American Society of Anesthesiologists (ASA) wellness score plus an additional time factor in 15-minute blocks multiplied by a conversion factor (USD 85.00). All base units from CPT codes for included procedures mapped to the same anesthesia Code (00145 = 6 base units). We initially ran independent-samples t-tests to determine if there were differences in costs between GA and LA categories. Our primary analysis included running three covariate-adjusted multivariate regression models with costs (OR, anesthesia, and PACU) as the dependent variable and anesthesia type (GA vs. LA) as the independent variable while adjusting for demographics (i.e., age categories, gender, and race), surgery type, smoking status, BMI, ASA score, and comorbidity score. Comorbidities were included but were not limited to chronic pulmonary disease, congestive heart failure, diabetes, hypertension, liver disease, cancer, renal failure, and alcohol abuse. Age was categorized by decade (e.g., <30 years, 30–39, 40–49, etc.) to allow for the capture of the expected complex, nonlinear relationship between GA use and age (i.e., young, and very old patients may be operated more under GA). All significance tests were two-tailed with an alpha of <0.05 to establish statistical significance. Stata 14.0 (StataCorp, College Station, TX, USA) was utilized for the analysis of data.

## 3. Results

Our analysis included 1473 eyes that received vitreoretinal surgery between 1 January 2015 and 31 September 2021. Of these surgeries, 635 (43.1%) received LA, while 838 (56.9%) received GA. During the selected period, there was an increasing trend of LA use from 2015–2019 (31.69%, 39.25%, 44.66%, 42.73%, and 50.85%, respectively) and then a reversal from 2020–2021 (44.66% and 42.61%) (Figure 1). Table 1 provides the demographic characteristics, risk factors, and durations (PACU, OR, and anesthesia) between the groups of patients undergoing general anesthesia and local anesthesia.

Divided by the type of vitreoretinal surgery, the number of eyes in the complex PPV group, noncomplex PPV group, PPV for retinal detachment, scleral buckle for retinal detachment, and the combined PPV group for retinal detachment were 323 eyes, 954, 167, 15 and 17 eyes, respectively (Figure 2). After logistical regression analysis, we found that complex pars plana vitrectomy and combined pars plana vitrectomy with scleral buckle predicted the use of GA (Odd Ratio (OR) = 2.41, 95% Confidence Interval (CI) 1.73–3.35, *p* < 0.001; OR = 44.18, 95% CI 2.53–76.90, *p* = 0.009, respectively). Current smoking and increased systemic comorbidities decrease the likelihood of using GA (OR = 0.94, 95% CI 0.93–0.95, *p* < 0.001; OR = 0.93, 95% CI 0.90–0.97, *p* = 0.001, respectively). Regarding the effect of age on the model outcome, we found a decreasing trend in the odds of using GA in patients with age brackets of 40-49 years and older (Table 2).

The mean (SD) surgical duration in minutes was 66.47 (34.86) and 100.14 (50.69) for local and general anesthesia, respectively. The mean (SD) anesthesia duration was 100.49 (38.05) for local anesthesia as compared to 145.53 (53.92) for general anesthesia. PACU duration was found to be 53.38 (22.92) and 75.08 (34.82) for local and general anesthesia, respectively (Figure 3). We found a lower total hospital stay of 100.41 minutes when patients received LA compared to GA for vitreoretinal surgeries.

We found in a univariate analysis that the average cost (SD) of OR time for the LA group was USD 5393 (USD 1532) lower compared to USD 7236 (USD 2203) in the GA group, with a mean difference (SEM) of USD 1843 (USD 102.20). The cost of anesthesia was also lower for surgeries performed with LA (USD 1141 ± USD 227) compared to those under GA (USD 1383 ± USD 307), with a mean difference of USD 242 ± USD 14.49 (*p* < 0.001). The cost of PACU time for those with LA was found to be USD 886 (USD 130) in the LA group, while in the GA, the cost of PACU time was found to be USD 1000 (USD 260), a mean difference of USD 113 ± USD 11.30 (*p* < 0.001) (Figure 4). In total, the cost of surgery under LA was lower than GA by approximately USD 2199, a cost reduction of 22.86%.

We fitted three multivariate regression models to determine the increased costs associated with general anesthesia in the OR, anesthesia, and PACU while adjusting for patient characteristics, surgery type/complexity, and other risk factors that may increase the use of GA (Table 3). We found that the use of LA leads to a USD 173.41 (95% CI, USD 142.91–USD 203.90) decrease in anesthesia cost over GA (*p* < 0.001). The use of LA was found to lead to a USD 90.90 (95% CI, USD 65.46–USD 116.34) decrease in PACU cost. The greatest reduction in price was found to be OR costs, with LA leading to a USD 1251.32 (95% CI, USD 1034.44–USD 1468.20) decrease in cost over GA. The total adjusted cost of surgery under LA was lower than GA by approximately USD 1515.63, a cost reduction of 15.76%.

## 4. Discussion

In this single-site study, we analyzed the operational efficiency and cost differences between local and general anesthesia for vitreoretinal surgery anesthesia. We found that surgical time, anesthesia time, and PACU duration were significantly lower in the LA group by approximately 100.41 minutes per surgery. The average cost of OR, anesthesia, and PACU times were substantially lower for the LA group by USD 2199 per surgery compared to the GA group, a 22.86% cost reduction. On adjusted analysis, controlling for covariates that may influence the choice of anesthesia delivery such as patients’ age, complexity of surgery, smoking, and comorbidity score, we found that cost differences were slightly lower, with a total cost reduction of USD 1515.63, a 15.76% cost reduction.

With the rising cost of health care, it is imperative to analyze the operational efficiency and cost-reduction strategies of surgery facilities. The costs of the operating room and postoperative anesthesia care unit (PACU) largely depend on the time spent in each area. The time spent in each area also determines the equipment and staffing necessary for each patient. Two essential surgeon-controlled factors include surgical time and the type of anesthesia delivered [8]. We found that patients in the LA group had shorter surgical duration, anesthesia time, and PACU stays prior to discharge from the hospital compared to the GA group (66.47 min vs. 100.14 min, 100.49 min vs. 145.53 min, and 53.38 min vs. 75.08 min, respectively). This led to a total saving of approximately 100.41 min and a rapid turnover of patients in both the OR and the PACU. Based on the set tariff for and after accounting for other covariates that can influence the choice of anesthesia, such as surgery complexity and patients’ comorbidities, we found the difference in cost of OR, anesthesia, and PACU times to be lower in LA by USD 1251.32, USD 173.41, and USD 90.90, respectively, with an estimated total cost reduction per surgery of about USD 1516. Although our study showed an adjusted 15.76% (USD 1515.63) cost reduction per procedure using LA, we did not account for the anesthetic medication cost in our study. Because medications used in GA are in general more expensive than those used for LA, we expect the cost reduction with LA to be even higher. In a small study of 100 procedures comparing LA vs. GA for vitreoretinal surgery in Indonesia, Simanjuntak et al. found a total cost savings of 46.06% with the use of LA over GA with the inclusion of anesthetic medication usage [8]. It is of note that the cost savings may vary by institution depending on the used tariff.

There is enhanced operational efficiency with LA use for surgery facilities, with a lower cost burden for patients and surgical facilities. As demonstrated in our study, local anesthesia for vitreoretinal surgery has proven advantages to patients, such as quicker recovery and earlier discharge from the surgery facility. Additionally, LA is less associated with cardiovascular risk, particularly for patients with coexisting comorbidities [3,8]. However, because variables such as a patient’s age, medical comorbidities, surgical complexity, and expected case time may also play a role in the choice of anesthesia type [11] it was important to account for the effect of those factors on the estimated cost reduction, as demonstrated in this study.

In this study, 38.2% of patients received LA and 61.8% received GA for vitreoretinal surgery. Our data support the prediction of GA for younger patients <40 years of age and for complex vitreoretinal surgery. The use of GA in younger patients may be attributable to the likelihood of choosing to perform a scleral buckle procedure in this patient cohort because of anatomical factors such as lack of posterior vitreous detachment and phakic status. A scleral buckle can be a more painful procedure because of the traction with the handling of extraocular muscles and cryopexy compared to a stand-alone simple pars plana vitrectomy. Additionally, complex PPV and combined PPV and scleral buckles are expected to be prolonged, leading to an increased chance of GA use. Our data also demonstrate that from 2015 to 2019 there was an increasing trend of using LA over GA. From 2020 to 2021, there was a small reversal of this trend, which may be attributable to elective cases at our institution being canceled or postponed because of the international COVID-SARS-2 pandemic. We infer that those cases during this time period were more likely to be urgent and more complex in our tertiary referral center, likely leading to the choice of GA over LA (Table 2).

There are limitations to this study. First, the study is limited by its retrospective, nonrandomized design, which may introduce outcome and selection bias. We did not analyze surgical success or failure rates between the two groups in this study. Additionally, there may be an inherent selection bias from 2020 to 2021 during the pandemic’s closure of elective cases toward the use of GA. In addition, patients in this study may also have more complex vitreoretinal pathologies than those presenting to non-tertiary-care centers where the case mix will be skewed more toward macular, noncomplex vitreoretinal surgery. Furthermore, it is of note that our calculations did not account for medicine or supply expenses and was totally based on the cost of anesthesia, operation room, and perioperative area utilization. General anesthesia drugs have higher acquisition costs than local anesthesia, though the negotiated reimbursement plans for surgery between surgery facilities and insurance companies do not usually differentiate between surgeries performed under local or general anesthesia. We did not analyze cost drivers for MAC and GA use. An important future course would be to analyze the cost drivers in these groups. Finally, given that patients receiving LA receive minimum to no sedation, these patients typically need less supervision by the anesthesia providers after surgery as compared to GA. However, we were not able to measure this variable in the current study.

Our study has several strengths. Prior to our report, no contemporary studies have investigated the cost of different types of vitreoretinal surgery anesthesia in Western countries. In addition to presenting the expected benefits in time and cost reduction in using LA, we also provided an adjusted estimate to account for the effect of factors that necessitate the use of GA, such as a young patient’s age and complex retina surgery. Our data are not only relevant to the Western world but can also be helpful to developing countries with limited resources. Our results should aid surgeons and anesthesiologists to understand the operational efficiency and cost effectiveness of anesthesia delivery methods for vitreoretinal surgery.

## 5. Conclusions

In summary, we provide data on the time-saving and cost differences of the use of LA over GA in the outpatient setting. We found that anesthesia, surgical, and recovery times were significantly lower for eyes that received local anesthesia, and this was associated with an enhanced operational efficiency, rapid turnover, and a cost-reduction estimate of USD 1516 per surgery with using LA. This is a conservative estimate, and we expect the cost savings to increase if the cost of anesthetic medications is also accounted for.

## Figures and Tables

**Figure 1 healthcare-10-01918-f001:**
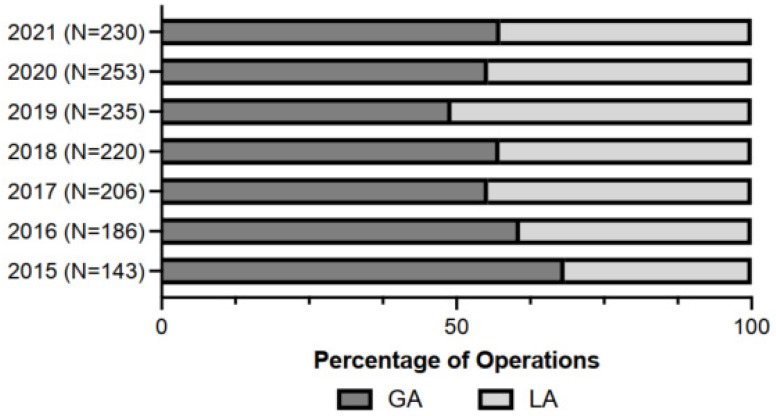
Graph showing patterns of anesthesia used for vitreoretinal surgery during the period between 2015 and 2021.

**Figure 2 healthcare-10-01918-f002:**
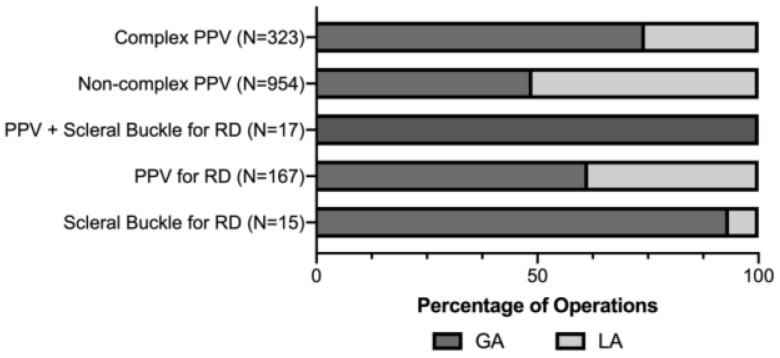
Graph showing the type of anesthesia used according to the complexity of the vitreoretinal surgery.

**Figure 3 healthcare-10-01918-f003:**
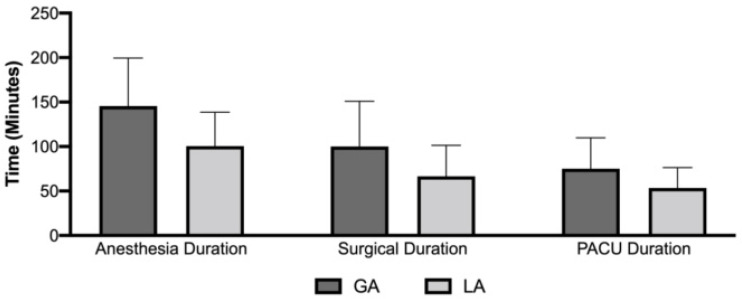
Graph comparing the anesthesia and surgery duration under general anesthesia (GA) compared to local anesthesia (LA). The anesthesia duration and the surgery duration were significantly higher in the GA group.

**Figure 4 healthcare-10-01918-f004:**
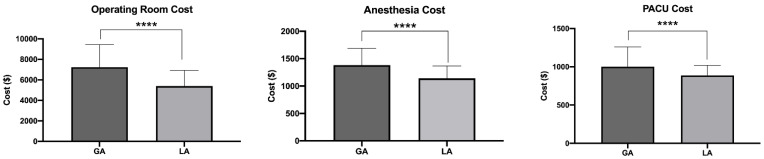
Costs for OR, anesthesia, and PACU between GA and LA groups, **** *p* < 0.0001.

**Table 1 healthcare-10-01918-t001:** Demographic Characteristic, Procedural Duration, Anesthesia Duration, and PACU Duration Between General Anesthesia and Non-General Anesthesia Groups Undergoing Vitreoretinal Surgery.

	General Anesthesia	Local Anesthesia
**Total, No. (%)**	838 (56.89)	635 (43.11)
**Demographics**		
Gender, No. (%)		
Women	372 (54.39)	312 (45.61)
Men	466 (59.06)	323 (40.94)
Age Categories, No. (%)		
<30	88 (93.62)	6 (6.38)
30–39	89 (89.00)	11 (11.00)
40–49	132 (80.98)	31 (19.02)
50–59	171 (65.52)	90 (34.48)
60–69	193 (52.59)	174 (47.41)
70–79	121 (35.38)	221 (64.62)
80–89	36 (31.03)	80 (68.97)
>90	8 (26.67)	22 (73.33)
Race, No. (%)		
Black	266 (58.46)	189 (29.21)
White	492 (54.42)	412 (45.58)
Asian	7 (50.00)	7 (50.00)
Other	63 (70.79)	26 (29.21)
**Smoking Status, No. (%)**		
Never Smoked	425 (54.35)	357 (45.65)
Former Smoker	210 (52.90)	187 (47.10)
Current Smoker	190 (69.34)	84 (30.66)
**Comorbidity Score, mean (SD) ^a^**	3.62 (3.71)	4.79 (4.16)
**BMI, mean (SD)**	29.89 (7.39)	29.59 (6.29)
**Surgery Type, No. (%)**		
Noncomplex PPV	465 (48.84)	487 (51.16)
Scleral Buckle	14 (93.33)	1 (6.67)
PPV for RD	102 (61.45)	64 (38.55)
PPV + Scleral Buckle for RD	17 (100)	0 (0)
Complex PPV	240 (74.30)	83 (25.70)
**Procedure Duration, mean (SD)**	100.14 (50.69)	66.47 (34.86)
**Post-Anesthesia Care Unit, mean (SD)**	75.08 (34.82)	53.38 (22.92)
**Anesthesia Duration, mean (SD)**	145.53 (53.92)	100.49 (38.05)

Note. BMI, body mass index; SD, standard deviation, No., frequency. ^a^ Comorbidities included, but were not limited to, chronic pulmonary disease, congestive heart failure, diabetes, hypertension, liver disease, cancer, renal failure, and alcohol abuse.

**Table 2 healthcare-10-01918-t002:** Adjusted Odds Ratios for Performance of Vitreoretinal Surgery Type Under General Anesthesia.

General Anesthesia ^a^	Odds Ratio	95% CI	*p*-Value
**Surgery Type ^b^**			
Noncomplex PPV	*Reference*		
Scleral Buckle	3.73	0.59–23.43	0.160
PPV for RD	1.39	0.93–2.07	0.105
PPV + Scleral Buckle for RD	44.18	2.53–76.90	0.009
Complex PPV	2.41	1.73–3.35	<0.001
**Race**			
White	*Reference*		
Black	0.97	0.72–1.32	0.866
Asian or Pacific Island	1.86	0.61–5.64	0.276
Other	1.23	0.70–2.16	0.465
**Smoking Status**			
Non-Smoker	*Reference*		
Former Smoker	1.21	0.90–1.62	0.204
Current Smoker	1.76	1.24–2.51	0.002
**Age**	0.94	0.93–0.95	<0.001
<30		*Reference*	
30–39	0.65	0.21–2.02	0.459
40–49	0.34	0.13–0.91	0.032
50–59	0.17	0.07–0.43	<0.001
60–69	0.10	0.41–0.26	<0.001
70–79	0.05	0.02–0.13	<0.001
80–89	0.05	0.02–0.14	<0.001
>90	0.05	0.15–0.18	<0.001
**Sex**	1.12	0.87–1.45	0.381
**BMI**	1.02	0.99–1.04	0.137
**Comorbidity Score ^c^**	0.93	0.90–0.97	0.001

Note. BMI, body mass index; PPV, pars plana vitrectomy; RD, retinal detachment. ^a^ Logistic regression model with performance under general anesthesia as a dependent variable adjusting for age, sex, race, smoking, BMI, surgery type, and comorbidities. ^b^ PPV for RD included cases of primary repair of retinal detachment with no scleral buckle mentioned in the surgery description column. ^c^ Comorbidities included, but were not limited to, chronic pulmonary disease, congestive heart failure, diabetes, hypertension, liver disease, cancer, renal failure, and alcohol abuse.

**Table 3 healthcare-10-01918-t003:** Adjusted Regression Models for Operating Room, PACU, and Anesthesia Cost by Use of General Anesthesia.

	Adjusted R^2^	B	Standard Error	95% CI	*p*-Values ^b^
**Costs ^a^**					
Operating room	0.326	1251.32	110.56	1034.44–1468.20	<0.001
PACU	0.066	90.90	1.0034	65.46–116.34	<0.001
Anesthesia	0.313	173.41	15.54	142.91–203.90	<0.001

Notes. OR, odds ratio; PACU, post-anesthesia care unit; CI, confidence interval. ^a^ Three confounder adjusted logistic regression models were performed with costs as the dependent variable and anesthesia type as the independent variable. ^b^ Adjusted for demographics (i.e., age categories, gender, race), surgery type, smoking status, BMI, American Society of Anesthesiologists (ASA) wellness score, and comorbidity score.

## Data Availability

The datasets generated during and/or analyzed during the current study are available from the corresponding author on reasonable request.

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
