# Peer review of "Analysis of Operational Efficiency and Cost Differences between Local and General Anesthesia for Vitreoretinal Surgery"

_healthcare, 2022, doi:10.3390/healthcare10101918_

Round 1

Reviewer 1 Report

1.       Introduction:

Justification why the authors need to assess “institutional benefit” of using LA vs GA for VR surgery is not clearly mentioned.what is incidence of retinal detachment in the USA?  It is decribed that number of VR procedures in US has increased and mostly on OP surgery basis. It is also mentioned that increased number of LA for VA surgery in Europe and the USA. There are various advantages of using LA over GA. So its clear that LA is prefereable. However, authors did not explain why cost matters, is there an complaint from patient or insurer in regard to high cost of GA eg claims or out-of-pocket payment or any other reasons?What do you mean by “institutional benefit” and “operational efficiency”? Any connection with hospital inefficiency in delivering services?

2.       Methods

The authors described how they extracted data retrospectively from EMR system of UAMS, a single-site tertiary care facility, covering a large data cohort from January 2015 to September 2021 (line 55-57). Please add informartion on:

-inclusion and exclusion criteria and come up with final data set of 1473 eyes with 635 receiving LA and 838 receiving GA (line 64-65). Usually patients with systemic diseases that may affect the choice of surgey were excluded in the analysis, how did you deal with this criteria?

-cost perspective (hospital?) and components (did you include all cost components ie fixed and variable costs, direct and indirect cost, overhead cost, maintenance cost etc or lab cost, surgery, medication?). You extracted cases from database, but you need to identify all relevant costs to the selected patients. How did you detemine the costs, did you work on the financial data or billing from hospital, or any other approach you used? Explain how did you combine it to come up with cost per patient. Or, did you apply a kind of acivity based or time-driven costing approach?

-other costs: you included comorbidity in your analysis, did you include cost of relevant treatments, including comorbidity or side effect?

-clinical outcome: although this study is a “partial economic evaluation”, focusing on cost analysis, you need to describe that the two groups were comparable. Any  intention to conduct CEA? The authors need to clarify that all selected patients from both group LA and GA were similar / comparable  in terms of clinical outcome  (visual acuity or other exam?)

-How did you measure “efficiency”?

3. Result:

-data inclued in the analysis were data from 2015-2021 , many factors could had happened / affected study result (other than pandemic) such as changing in policy, payment, insurance coverage, technology or even doctors and service availability etc. How did you manage these issues in your analysis?

-You mentioned that “the major outcomes of interest in this study were the operating room, anesthesia, and PACU times and the cost for vitreoretinal surgical procedures operated under local  anesthesia compared to general anesthesia”(line 80-81). How did you reflect those variables in your analysis? Which cost driver you found in your study?

-Line 119-122): Divided by the type of vitreoretinal surgery, the number of eyes in the complex PPV 119 group, noncomplex PPV group, PPV for retinal detachment, scleral buckle for retinal de- 120 tachment, and the combined PPV group for retinal detachment were 323 eyes, 954, 167, 15 121 and 17 eyes, respectively (Figure 2) . Please decribe how these type of VR surgery affected costs (providing table on this would be better)

-This is a cost analsis study, you have to describe systematically cost components especially related to the cost driver(s)

-You include comorbidity in the analysis, describe how comorbidity influence cost (its statistically significant, do you think cost to treat comorbid increased overall cost? How did you consider it in your inclusion exclusion criteria and cost determination. You cannot simply exclude in the cost but you include it in your statistical analysis

4. Discussion

-lack of description on how you achieved operational efficiency

-it is good to explain study limitation and stud strengths (Line 231-244) .....Patients in this study may also have more complex  vitreoretinal pathology than those presenting to non-tertiary care centers where the case  mix will be skewed more towards macular, noncomplex vitreoretinal surgery. Further, it is of note that our calculations did not account for medicine or supply expenses and was  totally based on the cost of anesthesia, operation room and perioperative area utilization.  General anesthesia drugs have higher acquisition costs than local anesthesia, though ne-gotiated reimbursement plans of surgery between surgery facilities and insurance companies do not usually differentiate if the surgery was performed under local or general anesthesia. Finally, given that patients receiving LA receive minimum to no sedation, these patients typically need less supervision by the anesthesia providers after surgery as compared to GA. However, we were not able to measure this variable in the current study...Yes you described it in the discussion that you were not able to measure this, to what extent this issue affect your study result?

5. Conslusion

-its is stated that authors provide data on the time-saving and cost-effectiveness of the use of LA over GA in the outpatient setting (line 253-254). Its is not clear how you could conlude as “cost-effective”, even “operational efficiency”.

-what would you recommend to improve efficiency based on the study result?

Author Response

   Introduction:

Justification why the authors need to assess “institutional benefit” of using LA vs GA for VR surgery is not clearly mentioned.what is incidence of retinal detachment in the USA?  It is decribed that number of VR procedures in US has increased and mostly on OP surgery basis. It is also mentioned that increased number of LA for VA surgery in Europe and the USA. There are various advantages of using LA over GA. So its clear that LA is prefereable. However, authors did not explain why cost matters, is there an complaint from patient or insurer in regard to high cost of GA eg claims or out-of-pocket payment or any other reasons? What do you mean by “institutional benefit” and “operational efficiency”? Any connection with hospital inefficiency in delivering services?

Response: Dear Editor, healthcare cost has been steadily rising in the USA, and we believe that due to the rising cost, we must identify solutions by cost utilization strategies. In vitreoretinal surgery, our study has found that by using local anesthesia over general anesthesia, we may be able to mitigate healthcare cost. Additionally, there is a significant decrease in time spent in the ambulatory surgery center for patients that receive local anesthesia over general anesthesia.

  1. Methods

The authors described how they extracted data retrospectively from EMR system of UAMS, a single-site tertiary care facility, covering a large data cohort from January 2015 to September 2021 (line 55-57). Please add informartion on:

-inclusion and exclusion criteria and come up with final data set of 1473 eyes with 635 receiving LA and 838 receiving GA (line 64-65). Usually, patients with systemic diseases that may affect the choice of surgery were excluded in the analysis, how did you deal with these criteria?

Response: Thank you for the suggestions and questions. Our primary analysis focused on understanding the cost differential between MAC and GA. As such, our inclusion criteria included all eyes that underwent VR surgery at our institution between January 2015 to September 2021 that fell within the 5 categories, based on the type of retina surgery (as determined by surgical CPT codes and ‘Procedure Text’ extract): Scleral buckle for retinal detachment (67107), pars plana vitrectomy (PPV) for retinal detachment (67108), PPV plus scleral buckle for retinal detachment (67108), Noncomplex PPV including macular surgery, vitreous hemorrhage and opacities, dropped nucleus (67041, 67042, 67043, 67036, 67015, 67030, 67031, 67039, 67040, 67121), and Complex PPV (67113). We subsequently filtered out multiple surgeries in each eye. Exclusion criteria included VR surgeries that did not fall within the previous categories, including the following CPT codes: 67005, 67010, 67027, 67028, 67145, 67141, 67220, 67208, 67210, 67221, 67225, 67227, 67228, 67229, 67550, 67515, 67025, 67101, 67105, 67120, 67250, 67225, 67115, and 67218.  We collected additional patient level covariates including demographics and pre-existing conditions. We did not exclude patients with systemic diseases from the analysis. Many patients we perform surgery on have systemic diseases. By removing these patients, we would not get the full picture of costs associated with MAC and GA use. Comorbidities included but were not limited to, chronic pulmonary disease, congestive heart failure, diabetes, hypertension, liver disease, cancer, renal failure, and alcohol abuse. We adjusted for comorbidities in the cost analysis.

-cost perspective (hospital?) and components (did you include all cost components ie fixed and variable costs, direct and indirect cost, overhead cost, maintenance cost etc or lab cost, surgery, medication?). You extracted cases from database, but you need to identify all relevant costs to the selected patients. How did you detemine the costs, did you work on the financial data or billing from hospital, or any other approach you used? Explain how did you combine it to come up with cost per patient. Or, did you apply a kind of acivity based or time-driven costing approach?

Response: This is an excellent question. Our calculations do not take into account all cost components. For example, we did not take into account any medication, supply charges, or indirect costs. We know that there would be slightly more acquired costs for general anesthesia medications vs MAC sedation medications based upon the aforementioned variables. We have considered these, but OR time and anesthesia time vastly occupy the main costs of the surgery. Our cost calculations are all based off of a base plus additional time charge. We have detailed this in the limitations section. We worked with the billing admin to calculate costs. The following are examples of our cost calculations:

Cost of operating room: (30 minute initial charge + [(x) of 15 minute blocks rounded down) x $616] $2,795 + (each additional 15 minute block is $616)

i.e. case lasting 120 minutes

(30 minute initial charge + [(90min/15min = 6 additional blocks of time x $616)] $2,795 +  $3,696 = $6,491 OR time

Cost of PACU time:(60 minute initial charge + [(x) of 15 minute blocks rounded down) x $125] $838.72 + [(x)  of 15 minute blocks rounded down x $125)]

i.e. 55 minute pacu stay

$838.72 + (0 x $125)  (because it less than 60 minutes) $838.25 PACU charge

Cost of anesthesia charges: (Note: No difference in anesthesia fees if the case is general or MAC. It’s all billed the same rate in our institute)

(Base Units+ Time Units+ Modifying Units) * Conversion Factor = charge for the anesthesia charges

CPT codes: 67107, 67108, 67113, 67041, 67042, 67043, 67036, 67015, 67030, 67031, 67039, 67040, 67121

These all map to the same Anesthesia Code: 00145. The base unit for these CPTs= 6 base units. Modifying units were based on the ASA score.

ASA 1 = 0 modifying unit

ASA 2 = 0 modifying unit

ASA 3 = 1 modifying unit

ASA 4 = 2 modifying units

(ASA 5 = death without the surgery, not relevant for these cases. ASA 6 = organ donors were not represented in the included procedures.})

i.e. General or MAC case that lasts 135 minutes for an ASA 2 patient

(Base Units+ Time Units+ Modifying Units) * Conversion Factor [(6 units + (135 min/15min = 9 units) +  0 units] * $20.30 15units * $20.30 = $304.50

We subsequently added up costs for each: OR + anesthesia + PACU.

-other costs: you included comorbidity in your analysis, did you include cost of relevant treatments, including comorbidity or side effect?

Response: We included the ASA score and comorbidities in the adjusted analysis. We did not include treatment costs.

-clinical outcome: although this study is a “partial economic evaluation”, focusing on cost analysis, you need to describe that the two groups were comparable. Any intention to conduct CEA? The authors need to clarify that all selected patients from both group LA and GA were similar / comparable  in terms of clinical outcome  (visual acuity or other exam?)

Response: This study evaluated operation room time between local and general anesthesia from a same-day outpatient surgery center. We did not evaluate clinical outcomes.

-How did you measure “efficiency”?

Response: Efficiency was measured in time-saving.

  1. Result:

-data inclued in the analysis were data from 2015-2021 , many factors could had happened / affected study result (other than pandemic) such as changing in policy, payment, insurance coverage, technology or even doctors and service availability etc. How did you manage these issues in your analysis?

Response: All patients were operated on by the same two vitreoretinal surgeons. Additionally, all patients had their surgery in the same one-day outpatient surgery facility at a tertiary referral center. Accounting for further confounders such as changes in policy, payment, and insurance coverage were not analyzed.

-You mentioned that “the major outcomes of interest in this study were the operating room, anesthesia, and PACU times and the cost for vitreoretinal surgical procedures operated under local  anesthesia compared to general anesthesia”(line 80-81). How did you reflect those variables in your analysis? Which cost driver you found in your study?

Response: Thank you for your question. These variables were the primary outcomes in our study. Thus they were reflected as the dependent variable in the regression models. Three confounder adjusted logistic regression models were performed with calculated costs for operating room, anesthesia, and PACU  as the dependent variable and anesthesia type as the independent variable. We adjusted for demographics (i.e., age categories, gender, race), surgery type, smoking status, BMI, American society of anesthesiologists (ASA) wellness score, and comorbidity score. The primary driver of cost for each was the actual time spent in each. We did not perform a sub analysis determining factors associated with increased time and subsequent cost for operating room, anesthesia, and PACU. This is outside scope, but would be an interesting follow-up study. We have included this as an important area of future exploration. Thank you for your suggestion.

-Line 119-122): Divided by the type of vitreoretinal surgery, the number of eyes in the complex PPV 119 group, noncomplex PPV group, PPV for retinal detachment, scleral buckle for retinal de- 120 tachment, and the combined PPV group for retinal detachment were 323 eyes, 954, 167, 15 121 and 17 eyes, respectively (Figure 2) . Please describe how these type of VR surgery affected costs (providing table on this would be better)

Response: This study utilized time in each ambulatory care setting (Surgery, Anesthesia, PACU) as surrogate estimates for cost values. Equipment costs between different surgical instruments or procedures was not analyzed and is outside the scope of this study.

-This is a cost analysis study, you have to describe systematically cost components especially related to the cost driver(s)

Response:  Thank you very much for this suggestion. We believe that an analysis of the cost drivers in each modality would be a very interesting follow-up study. However, this was not within the scope of this study. Previous work on cost drivers has shown that anesthesia costs are strongly predicted by differences in average case duration, with other known or presumed cost drivers adding little to explain the cost differences. Our analysis calculations focused centrally on OR, anesthesia, and PACU duration to calculate costs. We have included this as an important area of future research in the limitation section.

https://pubs.asahq.org/anesthesiology/article/101/6/1435/6709/Effect-of-Different-Cost-Drivers-on-Cost-per

-You include comorbidity in the analysis, describe how comorbidity influence cost (its statistically significant, do you think cost to treat comorbid increased overall cost? How did you consider it in your inclusion exclusion criteria and cost determination. You cannot simply exclude in the cost but you include it in your statistical analysis

Response: Thank you for your questions. We showed that comorbidity score predicted the use of GA, with patients with lower score more likely to use GA over MAC (Table 2). We also included comorbidity score in our adjusted analysis of cost (Table 3). We included all patients who underwent VR surgery as detailed in our inclusion criteria. We did not exclude based upon comorbidities, but instead adjusted for this in the analysis of costs between modalities across OR, anesthesia and PACU.

  1. Discussion

-lack of description on how you achieved operational efficiency

Response: We believe a patient spending less time in each facet of the ambulatory surgery center would entail operational efficiency. Overall, patients spent an average of ~100 minutes less in the ambulatory surgery center with the use of LA over GA.

-it is good to explain study limitation and stud strengths (Line 231-244) .....Patients in this study may also have more complex  vitreoretinal pathology than those presenting to non-tertiary care centers where the case  mix will be skewed more towards macular, noncomplex vitreoretinal surgery. Further, it is of note that our calculations did not account for medicine or supply expenses and was  totally based on the cost of anesthesia, operation room and perioperative area utilization.  General anesthesia drugs have higher acquisition costs than local anesthesia, though ne-gotiated reimbursement plans of surgery between surgery facilities and insurance companies do not usually differentiate if the surgery was performed under local or general anesthesia. Finally, given that patients receiving LA receive minimum to no sedation, these patients typically need less supervision by the anesthesia providers after surgery as compared to GA. However, we were not able to measure this variable in the current study...Yes you described it in the discussion that you were not able to measure this, to what extent this issue affect your study result?

Response: We utilized a modest cost model, and if medications used in general anesthesia were also included, we anticipate the cost difference and savings between local anesthesia and general anesthesia being even higher than what we calculated, given that medication costs are a large expense.

  1. Conslusion

-its is stated that authors provide data on the time-saving and cost-effectiveness of the use of LA over GA in the outpatient setting (line 253-254). Its is not clear how you could conlude as “cost-effective”, even “operational efficiency”.

Response: Thank you. The reviewer's point is well taken. We have removed the wording of cost-effective from the manuscript as this would need to be an additional analysis. For operational efficiency, our working definition is the measure of the proportion of costs incurred during the operation, and  anesthesia and PACU stays, where lower costs equate with greater efficiency.

-what would you recommend to improve efficiency based on the study result?

Response: This is a great question. Based on our findings, we believe that the use of MAC over GA would be more efficient in the setting of VR surgeries. Future studies should explore how to improve efficiency within each modality for VR surgeries.

Reviewer 2 Report

The paper is well written, but is at least 20 years delayed. Today general anesthesia in Ophthalmology is limited in very fez cases. All considerations in the manuscript sere acquired years ago. So the interest on this paper is very limited.

the Authors must try to highlight something different to incerasse readers interest

Author Response

The paper is well written, but is at least 20 years delayed. Today general anesthesia in Ophthalmology is limited in very fez cases. All considerations in the manuscript sere acquired years ago. So the interest on this paper is very limited.

Response: We believe there is interest in this paper as the growing trend of using LA over GA has been more paramount over the last decade. This is the first paper to analyze the cost differences between the two modalities of anesthetic care. Additionally, we believe there is interest amongst readers given the unique data for vitreoretinal surgery and the trends in Ophthalmology and in other surgical subspecialties on performing more surgical cases with local anesthesia.

the Authors must try to highlight something different to increase readers interest

Response: This is the first study in the United States discussing the cost differences and time saved by choosing LA over GA.

Reviewer 3 Report

Congrats for your interesting article. But I think it could be improved as follows:

In methods, you may define the study design (descriptive analysis, I think), 

“An institutional board review exemption was issued by the university, and no patient consent was required” pag2,  58- 59

The first paragraph analyzes eyes and the third, patients: you must choose a variable (eyes or patients) and refer to it throughout the text.

Could you explain better how this was calculated  “We calculated the costs of the operating room and PACU with two different equations that included an initial base charge ($2,795.00 and $838.72, respectively) plus an additional time charge in 15-minute blocks ($616.00 and $125.00, respectively). (files 82-84)

How was calculated surgical time? Was anesthesia time included?

It must be explained which professional performs the anesthetic block (surgeon, anesthesiologist,...) or general anesthesia and the procedure used. As well as the criteria for home discharge. Both variables could explain the disproportionate difference in costs.

In results: Please, review the figures: some errors have been detected

Discussion, should be developed following the results obtained, their congruence with the literature and the possible causes of it. Please, avoid large paragraphs, as well

Author Response

Congrats for your interesting article. But I think it could be improved as follows:

In methods, you may define the study design (descriptive analysis, I think),

Response: Thank you for your comment. Yes, in the methods we detail our study design stating our data collection, primary outcomes of interest, and statistical methods. Descripives are provided in Table 1.

“An institutional board review exemption was issued by the university, and no patient consent was required” pag2,  58- 59

Response: Thank you for your comment. We received an institutional board review exemption for this study.

The first paragraph analyzes eyes and the third, patients: you must choose a variable (eyes or patients) and refer to it throughout the text.

Response: Thank you for catching this. We have updated methods to clarify that we are talking about eyes as the primary unit of analysis.

Could you explain better how this was calculated  “We calculated the costs of the operating room and PACU with two different equations that included an initial base charge ($2,795.00 and $838.72, respectively) plus an additional time charge in 15-minute blocks ($616.00 and $125.00, respectively). (files 82-84)

Response: Thank you for your question Here are examples:

Cost of operating room: (30 minute initial charge + [(x) of 15 minute blocks rounded down) x $616] $2,795 + (each additional 15 minute block is $616)

i.e. case lasting 120 minutes

(30 minute initial charge + [(90min/15min = 6 additional blocks of time x $616)] $2,795 +  $3,696 = $6,491 OR time

Cost of PACU time:(60 minute initial charge + [(x) of 15 minute blocks rounded down) x $125] $838.72 + [(x)  of 15 minute blocks rounded down x $125)]

i.e. 55 minute pacu stay

$838.72 + (0 x $125)  (because it less than 60 minutes) $838.25 PACU charge

Cost of anesthesia charges: (Note: No difference in anesthesia fees if the case is general or MAC. It’s all billed the same rate in our institute)

(Base Units+ Time Units+ Modifying Units) * Conversion Factor = charge for the anesthesia charges

CPT codes: 67107, 67108, 67113, 67041, 67042, 67043, 67036, 67015, 67030, 67031, 67039, 67040, 67121

These all map to the same Anesthesia Code: 00145. The base unit for these CPTs= 6 base units. Modifying units were based on the ASA score.

ASA 1 = 0 modifying unit

ASA 2 = 0 modifying unit

ASA 3 = 1 modifying unit

ASA 4 = 2 modifying units

(ASA 5 = death without the surgery, not relevant for these cases. ASA 6 = organ donors were not represented in the included procedures.})

i.e. General or MAC case that lasts 135 minutes for an ASA 2 patient

(Base Units+ Time Units+ Modifying Units) * Conversion Factor [(6 units + (135 min/15min = 9 units) +  0 units] * $20.30 15units * $20.30 = $304.50

We subsequently added up costs for each: OR + anesthesia + PACU.

How was calculated surgical time? Was anesthesia time included?

 Response: Thank you for your comment. We have included in the text the way surgical time and anesthesia time were calculated. Briefly, Anesthesia time starts when an anesthesia provider enters the operating room with the patient.  Anesthesia stop takes place when the anesthesia provider delivers the patient to PACU and concludes the handoff to the recovery room nurse. Surgery time starts when the ophthalmologist makes the initial invasive incision and is communicated to the operating room circulator who documents procedure start in the EMR. Surgery time ends when the ophthalmologist has made their final closure/bandage of the procedure site, and this is documented by the operating room circulator.

It must be explained which professional performs the anesthetic block (surgeon, anesthesiologist,...) or general anesthesia and the procedure used. As well as the criteria for home discharge. Both variables could explain the disproportionate difference in costs.

Response: Thank you for your comment. At our institution, the Ophthalmologist performs the anesthetic block. The anesthesiologist or CRNA under the guidance of the anesthesiologist performs the endotracheal intubation for general anesthesia. Home discharge is determined by hemodynamic stability in the PACU and at the discretion of the anesthesiologist.

In results: Please, review the figures: some errors have been detected

Response: Thank you for the comments. We have reviewed the figures and we have not identified errors. Can you please clarify, so that we can address those?

Discussion should be developed following the results obtained, their congruence with the literature and the possible causes of it. Please, avoid large paragraphs, as well

Response: We have added all the relevant literature. There is a paucity of literature with the trend of local versus general anesthesia in vitreoretinal surgery. We included the largest study, which was published form the United Kingdom. We included a cost analysis from Indonesia, the only other study to address the cost differences between local versus general anesthesia in vitreoretinal surgery.